# Risk Factors for Postoperative Pulmonary Complications Leading to Increased In-Hospital Mortality in Patients Undergoing Thoracotomy for Primary Lung Cancer Resection: A Multicentre Retrospective Cohort Study of the German Thorax Registry

**DOI:** 10.3390/jcm11195774

**Published:** 2022-09-29

**Authors:** Wolfgang Baar, Axel Semmelmann, Julian Knoerlein, Frederike Weber, Sebastian Heinrich, Torsten Loop

**Affiliations:** 1Department of Anaesthesiology and Critical Care, Medical Center—University of Freiburg, Faculty of Medicine—University of Freiburg, 79106 Freiburg, Germany; 2German Society of Anaesthesiology and Intensive Care Medicine, 10115 Nürnberg, Germany

**Keywords:** post-operative pulmonary complications, thoracotomy, lung cancer resection, in-hospital mortality of thoracotomy

## Abstract

Postoperative pulmonary complications (PPCs) represent the most frequent complications after lung surgery, and they increase postoperative mortality. This study investigated the incidence of PPCs, in-hospital mortality rate, and risk factors leading to PPCs in patients undergoing open thoracotomy lung resections (OTLRs) for primary lung cancer. The data from 1426 patients in this multicentre retrospective study were extracted from the German Thorax Registry and presented after univariate and multivariate statistical processing. A total of 472 patients showed at least one PPC. The presence of two PPCs was associated with a significantly increased mortality rate of 7% (*p* < 0.001) compared to that of patients without or with a single PPC. Three or more PPCs increased the mortality rate to 33% (*p* < 0.001). Multivariate stepwise logistic regression analysis revealed male gender (OR 1.4), age > 60 years (OR 1.8), and current or previous smoking (OR 1.6), while the pre-operative risk factors were still CRP levels > 3 mg/dl (OR 1.7) and FEV1 < 60% (OR 1.4). Procedural independent risk factors for PPCs were: duration of surgery exceeding 195 min (OR 1.6), the amount of intraoperative blood loss (OR 1.6), partial ligation of the pulmonary artery (OR 1.5), continuing invasive ventilation after surgery (OR 2.9), and infusion of intraoperative crystalloids exceeding 6 mL/kg/h (OR 1.9). The incidence of PPCs was significantly lower in patients with continuous epidural or paravertebral analgesia (OR 0.7). Optimising perioperative management by implementing continuous neuroaxial techniques and optimised fluid therapy may reduce the incidence of PPCs and associated mortality.

## 1. Introduction

Despite the decreasing lung cancer mortality rates in Europe and the USA, bronchial carcinoma remains one of the most frequently diagnosed malignancies, with a high mortality [1]. Surgery is a possible curative approach for treating early-stage lung cancer. Therefore, the need for these procedures is rising [2]. Patients presenting for lung surgery are often of older age and suffer from numerous comorbidities, leading to various complications and unplanned admission into ICUs with subsequent increased mortality and length of hospital stays [3]. Postoperative pulmonary complications (PPCs) following lung surgery are very frequent, with an incidence of up to 24%, and they cause up to 68% of unplanned ICU admissions [3,4]. When the resulting respiratory failure necessitates non-invasive ventilation and/or endotracheal re-intubation with prolonged mechanical ventilation after lung surgery, short- and long-term mortality is increased [5]. Some of the underlying risk factors leading to an increased rate of PPCs have been examined in the past. Increased age, body mass index (BMI) ≥ 30 kg/m^2^, pre-existing obstructive lung disease, and fluid overload are generally accepted risk factors for PPCs [6,7,8,9].

In a multicentre retrospective register study that addressed various risk factors for PPCs after video-assisted thoracoscopic (VATS) lung resections, we found duration of surgery and intraoperatively administered crystalloids exceeding 6 mL/kg/h as modifiable risk factors for PPCs [10]. As this study only included VATS lung resections, we performed a further multicentre retrospective analysis to identify risk factors for PPCs in patients undergoing open thoracotomy lung resections (OTLRs). The underlying question of this study was whether anaesthesia-related factors have a significant effect on the incidence of PPCs in OTLR. This study provides three major contributions: Firstly, it provides an up-to-date evaluation of the incidence of PPCs after OTLRs, including the baseline characteristics of patients who developed PPCs and those who did not. Secondly, it assesses the burden of disease for patients suffering from PPCs after OTLRs regarding mortality and length of ICU stay. Thirdly, the risk factors associated with patients, surgery, and anaesthesia that lead to PPCs are evaluated.

## 2. Materials and Methods

### 2.1. Data Source

Data were obtained from a multicentre retrospective cohort register and database of patients undergoing non-cardiac thoracic surgery and receiving an open thoracotomy during the calendar years 2016 to 2020. Therefore, the data were identified and extracted from the 2021 database of the German Thorax Registry, resulting in 6917 patients from 4 contributing centres who received non-cardiac thoracic surgery. The contributing centres were the University Hospital of Freiburg, the Hospital of the University Witten/Herdecke-Cologne, and the University Hospitals of Düsseldorf and Munich. The German Thorax Registry is an interdisciplinary and multicentre database initiated by the German Society of Anaesthesiology and Intensive Care Medicine (DGAI) and the German Society of Thoracic Surgery (DGT) [11]. The participating hospitals submit their data to a central database through a web-based application. The data are pseudonymised, meaning that the patient’s name is replaced with a sequence of numbers. The present study is in line with the publication guidelines of the German Thorax Registry, and the application for data analysis was formally accepted and registered by its advisory board. The study was planned and designed in accordance with the initiative for strengthening the reporting of observational studies in epidemiology (STROBE) by using the suggested checklist for epidemiological cohort studies (Appendix A) [12].

### 2.2. Ethics

Ethical approval for this study (Ethical Committee of the University Witten/Herdecke; Approval No.: 64/2014) was provided by the Ethical Committee of the University Witten/Herdecke, Witten, Germany (Chairperson Prof P. W. Gaidzik) on 24 June 2014.

### 2.3. Patients

Figure 1 shows the selection of cases for this study. Only patients undergoing thoracotomy for noncardiac surgery were included. This query resulted in 2794 patients. After that, the patients were divided into two groups according to if they received an open thoracotomy and lung surgery for primary lung cancer (1426 patients) or indications other than primary lung cancer (1368 patients), such as lung metastasis of other cancer entities, benign tumours, or mediastinal tumours. 

All patients underwent at least general anaesthesia. One-lung ventilation (OLV) was established during open thoracotomy. The main interest was set on PPCs. PPCs were defined as a composite outcome with prolonged air leakage ≥7 days after surgery, postoperative pneumonia, the need for non-invasive ventilation (NIV) due to respiratory failure, a new chest drain after surgery, the requirement of re-intubation due to respiratory failure or continued ventilation after surgery, pleural empyema, or the requirement of extra corporal membrane oxygenation (ECMO). In our study, pneumonia was diagnosed according to the definition of the European perioperative clinical outcome (EPCO): new pulmonary infiltrate with associated leukocytosis, fever, new purulent sputum, need for antibiotic therapy, and increased oxygen demand via a face mask [13].

As demonstrated in Figure 1, significantly more PPCs were observed in patients receiving lung resections for primary lung cancer compared to patients without primary lung cancer. Therefore, a further statistical analysis was performed in this subgroup.

### 2.4. Parameters

Listed below are the patient-specific and preoperative characteristics that were included in the following univariate analysis: gender, body mass index (BMI), age, ASA score, smoking, pre-operative pulmonary infection in the 4 weeks prior to surgery, FEV_1_ ≤ 60%, pre-operative lung surgery, pre-operative radio- and/or chemotherapy (RCT), TNM staging, pre-operative haemoglobin level, pre-operative C- reactive protein level >3 mg/dl, and pre-operative blood gas analysis data.

The procedural characteristics included surgery- and anaesthesia-related parameters. Surgery-related characteristics included duration of surgery ≥195 min (75% quartile), the amount of resected lung tissue, and the total amount of intraoperative blood loss.

Anaesthetic management was analysed according to the specific anaesthesia techniques (single-shot intercostal block (ICB), single-shot paravertebral block (PVB), continuous neuraxial techniques: paravertebral catheter (PVC), thoracic epidural anaesthesia via catheter (TEA)), maintenance of anaesthesia with inhaled anaesthetics or as total intravenous anaesthesia, fluid and vasopressor therapy (intraoperative crystalloid infusion rate ≥6 mL/kg/h, total amount of intraoperative crystalloids, colloids, packed red blood cells (PRBCs), and fresh frozen plasma, and the need for intraoperative vasopressor therapy. Furthermore, the ventilator settings during one-lung ventilation, including the average positive end-expiratory pressure (PEEP) and the lowest inspiratory fraction of oxygen (FiO_2_), were analysed. The amount of fluid infusion was left to the anaesthesiologist in charge. In addition, all centres strove to use protective ventilation. An additional analysis regarded the need for partial ligation of the pulmonary artery and the need for continued invasive ventilation immediately after surgery.

### 2.5. Statistics

Univariate statistical analysis was performed by dividing the cohort into 2 patient groups that were with and without PPCs. Statistical analyses of the continuous variables were calculated using the Mann–Whitney U test. Categorical variables were analysed with the x^2^ test. For the multivariate stepwise logistic regression analysis for assessing the impacts of different parameters on the incidence of PPCs, only parameters that showed a *p*-value < 0.005 in the univariate analysis were used. The amount of resected parenchyma was not selected for the multivariate regression analysis, as it was prearranged by the oncological needs and could not be influenced by the patients, surgeons, or anaesthesiologists. For the multivariate analysis, the stepwise forward approach was applied, and the following parameters were included in the analysis: male gender; ≥60 years of age; current or previously smoking; CRP levels > 3 mg/dl; preoperative FEV_1_ < 60%; duration of surgery ≥ 195 min; amount of intraoperative blood loss per 100 mL; partial ligation of the pulmonary artery; transfusion of PRBC; OLV PEEP ≤ 5 cmH_2_O; OLV duration ≥ 175 min; continuous epidural or paravertebral analgesia; intraoperative crystalloids ≥ 6 mL/kg/h; mechanically ventilated admission to intensive care unit after surgery.

A *p*-value of ≤ 0.05 was considered statistically significant. IBM SPSS Statistics for Windows (Version 23.0 Armonk, NY, USA: IMB Corp.) was used for the statistical analysis.

## 3. Results

The data from 2,794 complete cases of patients undergoing OTLR revealed 750 cases in which at least one PPC was documented (27%). Of the 1426 patients who underwent OTLR for primary lung cancer, 472 (33%) had one or more PPCs. In the subgroup of patients with primary lung cancer, overall, 660 PPCs were identified; their specific incidence is shown in Figure 2.

Table 1 shows the results of the univariate analysis comparing the patient-specific pre-operative characteristics of patients who underwent OTLR for primary lung cancer resection with and without PPCs. Table 2 shows the procedure-related risk factors categorised by surgery- and anaesthesia-related characteristics of patients who underwent OTLR for primary lung cancer resection.

In patients with a single PPC (*n* = 324), the in-hospital mortality was comparable to that of patients without PPCs (*n* = 955) (1.3% vs. 1.2%) (Figure 3). Two PPCs were documented in 117 patients; this subgroup had a significantly increased mortality of 7% compared to the subgroup of patients without or with a single PPC (*p* < 0.001). Three or more PPCs were documented in 33 patients, with a corresponding mortality of 33%, which was significantly increased compared to that of patients with two PPCs (*p* < 0.001). This subgroup of patients included 26 patients with three PPCs, of whom eight died (31%), as well as five patients with four PPCs, of whom two died (40%). Two patients had five PPCs, and one of these patients died (50%). In-hospital mortality was significantly higher in the group of patients with PPCs than in the group without (5% vs. 1%, *p* < 0.001). The overall mortality rate in our cohort was 2.5%.

Figure 4 shows the results of the multivariate stepwise logistic regression analysis. Significant patient-related risk factors in the multivariate logistic regression analysis were male sex, age over 60 years, and current or previous smoking (male sex: OR 1.4, 95% CI 1.0–1.8, *p* = 0.032; ≥60 years of age: OR 1.8, 95% CI 1.3–2.4, *p* = 0.000; current or previous smoking: OR 1.6, 95% CI 1.1–2.4, *p* = 0.019). The significant pre-operative risk factors were still CRP levels > 3 mg/dl and FEV_1_ < 60% (CRP > 3 mg/dl: OR 1.7, 95% CI 1.1–2.6, *p* = 0.015; FEV_1_ < 60%: OR 1.4, 95% CI 1.0–2.0, *p* = 0.042). 

The surgical risk factors of duration of surgery ≥195 min (OR 1.6, 95% CI 1.2–2.1, *p* = 0.002), amount of intraoperative blood loss per 100 mL (OR 1.6, 95% CI 1.2–2.1, *p* = 0.001), and partial ligation of the pulmonary artery (OR 1.5, 95% CI 1.0–2.1, *p* = 0.032) correlated significantly with the occurrence of PPCs.

Three anaesthesia-related risk factors showed a significant correlation with PPCs in the multivariate logistic regression analysis: admission as an endotracheally intubated patient to the intensive care unit after surgery (OR 2.9, 95% CI 1.5–5.3, *p* = 0.001), continuous epidural or paravertebral analgesia (OR 0.7, 95% CI 0.5–0.9, *p* = 0.007), and infusion of intraoperative crystalloids at a rate greater than 6 mL/kg/h (OR 1.9, 95% CI 1.4–2.8, *p* = 0.000). 

## 4. Discussion

The main results of this study could be summarised as follows. 

The multicentre registry analysis found an overall incidence of PPCs after OTLR of 27%, which is within the previously reported range from 5% to 55% [9,14,15,16]. The recent data from a large cohort of 1426 patients from 2016 to 2020 showed that the patients who underwent OTLR for primary lung cancer had a significantly increased rate of PPCs. One-third of the patients who underwent OTLR for primary lung cancer developed at least one PPC.

The burden of disease in terms of in-hospital mortality was limited in patients with a single PPC, but was statistically significant and clinically relevant, increasing from 1% to 7% in patients with two and as high as 33% in patients with three or more PPCs. As far as we know, this is the first study to show a significant association between the number of PPCs and the subsequent mortality. In the multivariate regression analysis of our study, the significant patient-related risk factors that were likely to lead to the development of PPCs were male sex, age greater than 60 years, current or past smoking, and FEV1 < 60%. This suggests a strong correlation between risk factors and complications, especially in high-risk patients, as well as their deleterious effects.

According to the multivariate logistic regression analysis, the procedure-related risk factors for PPCs were revealed to be three surgery-related and three anaesthesia-related risk factors for PPCs that were independent. The surgical risk factors were the amount of intraoperative blood loss, which is only partially modifiable, as well as a duration of surgery exceeding 195 min. Difficult and complicative surgeries can lead to a prolonged operative duration, which potentially contributes to the occurrence of PPCs. In a previously published study, a duration of VATS surgery of more than 120 min was found to be an independent risk factor for PPCs, analogously to the results of the present study in patients undergoing OTLR [10]. The underlying factors are most likely the same. Another independent surgery-related risk factor that emerged was partial ligation of a pulmonary artery. Partial pulmonary banding is an emergency procedure in order to mechanically reduce pulmonary shunting of the non-ventilated lung during OLV. If the underlying conditions necessitating this manoeuvre or the manoeuvre itself are responsible for the increase in PPCs is not clear and could not be determined from the data available to us.

The anaesthesia-related risk factors were postoperative intensive care unit ventilation after surgery, infusion of intraoperative crystalloid fluids of more than 6 mL/kg/h, and lack of continuous epidural or paravertebral analgesia.

The transfer of ventilated patients from the OR to the ICU can be for many different reasons. While factors such as postoperative residual curarisation, hypothermia, bed rest to ensure surgical outcomes, and upper airway diseases are frequently cited as reasons for such transfers, these do not apply specifically to thoracic surgery. As a conclusion from the present work, on-table extubation should be sought in all patients after OTLR if it is feasible and safe and if the neurologic status and cardiorespiratory conditions permit it.

Our study underlines the importance of homeostasis through adequate intraoperative haemodynamic management. Both hypovolemia, in the form of increased blood loss, and hypervolemia, in the form of crystalloid infusion rates greater than 6 mL/kg/h, lead to increased rates of PPCs. It has been adequately studied that increased administered fluid volume is an important modifiable procedure-related intraoperative risk factor for PPCs in lung surgery [9,15,17,18,19]. These studies were conducted in lung surgery patients undergoing thoracotomy, but also on mixed cohorts of patients undergoing VATS or thoracotomy. Arslantas et al. described an intraoperative infusion rate of greater than 6 mL/kg/h as a risk factor for PPCs when analysing patients undergoing VATS and thoracotomy [9,17]. This threshold was also found to be a significant risk factor in our previous study that looked at risk factors for PPCs after VATS lung cancer resections [10].

Epidural analgesia is the gold standard for patients undergoing OTLR, but paravertebral analgesia has also been shown in studies to have a comparable analgesic efficacy with reduced side effects [20,21,22,23,24,25,26]. Our data confirm that both analgesic methods—when used as a continuous method—result in a significantly lower incidence of PPCs compared to the absence of a continuous neuroaxial analgesic regime. Nevertheless, differences in perioperative practices between the contributing centres cannot be completely excluded and limit the interpretation in addition to the retrospective nature of this study.

In the entry of data records in the German Thoracic Register, there are only a few items on the settings of the ventilator for one-lung ventilation. Missing ventilation parameters are, e.g., the plateau pressure, driving pressure, or peak pressure. In agreement with other published studies, our data support the recommendation to adjust the PEEP individually [27]. The standard setting of a PEEP of 5 cmH_2_O or even lower was significantly associated with the occurrence of PPCs, but setting a PEEP level between 5 and 7 cmH_2_O showed no such association. This result could contribute to the ongoing discussion about optimising respiratory settings in thoracic anaesthesia [28]. Still, the complexity of the interactions of factors renders general recommendations a difficult matter.

The extent of lung parenchymal resection is known to be a risk factor for PPCs after OTLR [14,29]. In our study, we found lobectomies or bi-lobectomies to be associated with PPCs, but not pneumonectomies. Contrary to our expectations, we found fewer pneumonectomies in the group of patients with at least one PPC (5%) than in the group without PPCs (10%). Currently, the common expectation is that pneumonectomy leads to far more PPCs [16,29].

One reason for our result could be that the expectation of a worse outcome led to better medical care and, thus, less frequent PPCs. The preoperative evaluation and patient selection process for these patients facing a substantial perioperative risk could have resulted in an undetected selection bias, as patients who were deemed not fit enough, e.g., those with a high ASA status, were not treated with radical surgery, e.g., pneumonectomy, causing high-risk patients to be excluded, or those considered to be high-risk patients could have received intensified perioperative care. Notably, the overall incidence of continuous neuroaxial analgesic procedures and prolonged ICU stays with more optimised care could be responsible for the differences in outcomes. The overall incidence of continuous neuroaxial analgesic procedures in our cohort was 58% (832 of 1426). In patients undergoing pneumonectomy, we found an incidence of continuous neuroaxial analgesia of 76% (87 out of 114), which was significantly (*p* < 0.001) higher compared to that in the overall cohort.

Comparable factors could contribute to the finding that significantly more patients in the subgroup without PPCs had an ASA status ≥ 3 in the univariate analysis. Several publications have identified the ASA score as an independent predictor of PPCs after lung surgery, using an ASA score ≥ 3 as a threshold [14,30].

Our study has several limitations that restrict the interpretation of the results. The study has a retrospective character, which is why the results can only be interpreted as associations in the sense of generating a thesis. The interpretation of data from medical documents relies on exact and complete documentation. The multicentre character does not exclude variations in perioperative medical management, but the large cohort could mitigate these differences and further represent common medical practice, as management generally differs between the institutions. Although the main techniques are similar in all participating centres, there are still different anaesthetic and analgetic regimes; therefore, the heterogeneous approaches could also lead to an impact on PPCs. Further prospective randomised studies are needed to prove the causality of the associations shown in our study. Although the EPCO definition of postoperative pneumonia was adopted, the other PPCs were neither defined according to a standardised classification system nor adopted from the systemic classification of mortality and morbidity after thoracic surgery [13,31].

## 5. Conclusions

Despite the limitations mentioned above, some valid conclusions can be drawn from our study. First, there is a statistically significant and clinically relevant increase in mortality when patients experience two, three, or even more PPCs compared to that in patients with one or no PPCs. Some of the risk factors for PPCs found in our study, such as male sex or age, cannot be changed by patients, surgeons, or anaesthesiologists. However, on-table extubation (if possible), continuous neuroaxial analgesia, timing of surgery when patients are free of infection, and avoidance of crystalloid fluid intake greater than 6 mL/kg/h can be positively influenced in whole or in part by the treatment team of surgeons and anaesthesiologists. The burden of disease in patients undergoing OTLR for primary lung cancer is so high that additional PPCs negatively affect the outcomes. It is thus worthwhile to do everything that is possible to minimise the occurrence of PPCs in patients undergoing OTLR, as the corresponding burden of disease and the subsequent mortality increases with each PPC.

## Figures and Tables

**Figure 1 jcm-11-05774-f001:**
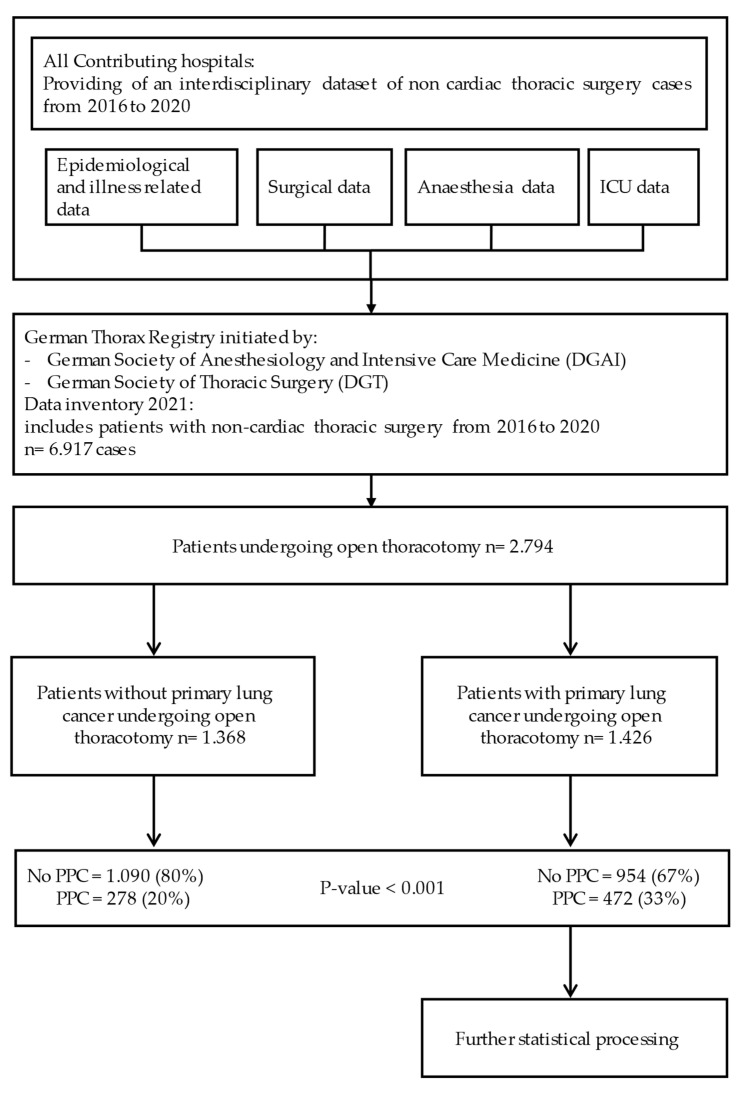
Case selection, study design, and entity-related PPC incidence. Data are presented as numbers of patients (percentage). PPC = postoperative pulmonary complications, ICU = intensive care unit, DGAI = German Society of Anaesthesiology and Intensive Care Medicine, DGT = German Society for Thoracic Surgery.

**Figure 2 jcm-11-05774-f002:**
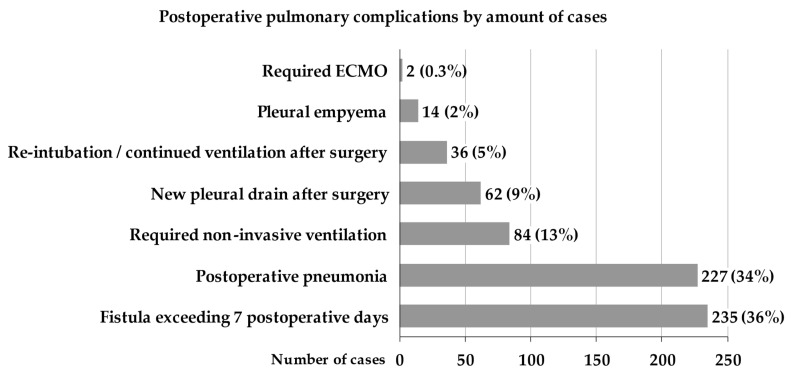
Postoperative pulmonary complications by number of cases. ECMO = extracorporeal membrane oxygenation.

**Figure 3 jcm-11-05774-f003:**
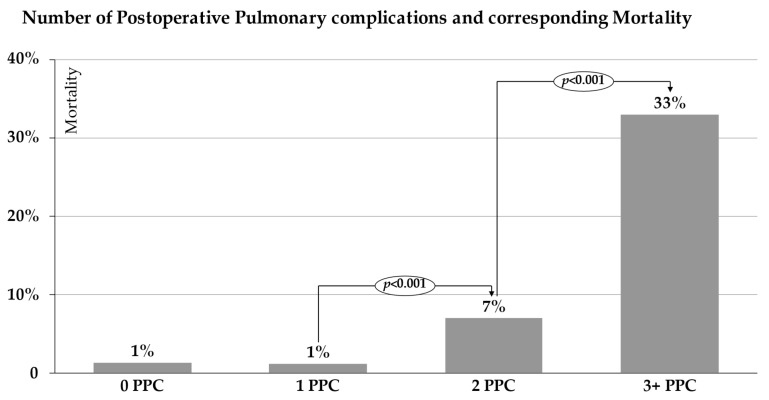
Number of postoperative pulmonary complications and corresponding mortality in percentages regarding cases with and without PPCs.

**Figure 4 jcm-11-05774-f004:**
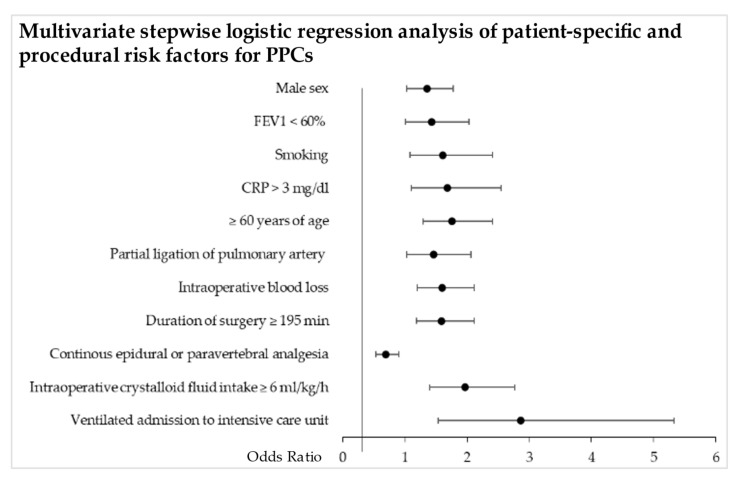
Multivariate stepwise logistic regression analysis of patient-specific and procedural risk factors for PPCs in primary lung cancer patients undergoing thoracotomy for lung resection. The OR and 95% CI are shown. CRP = C-reactive protein, FEV_1_ = forced expiratory volume in 1 s.

**Table 1 jcm-11-05774-t001:** Univariate analysis of patient-specific preoperative characteristics that led to risk factors for PPCs in patients undergoing open thoracotomy lung resection due to primary lung cancer. The data are presented as the number of patients (percentage) or median (±interquartile range). PPC = postoperative pulmonary complication; BMI = body mass index; CRP = C-reactive protein; ASA = American Society of Anesthesiology; FEV_1_ = forced expiratory volume in 1 s.

	No PPC (*n* = 954)	PPC (*n* = 472)	*p*-Value
**Patient-specific characteristics**			
Male gender	541 (57%)	315 (67%)	**<0.001**
Age [years]	65 ± 10	66 ± 9	0.001
≥60 years of age	678 (71%)	373 (79%)	**0.001**
BMI (mean +/−sd)	26.5 ± 5.8	26.4 ± 6.2	0.675
≤19	38 (4%)	22 (5%)	0.549
>30	201 (21%)	100 (21%)	0.959
Smoking			
never	165 (17%)	47 (10%)	**<0.001**
current	374 (39%)	209 (44%)	0.067
cessation ≥ 3 months	390 (41%)	206 (44%)	0.319
ASA ≥ 3	900 (94%)	431 (91%)	**0.031**
**Neoadjuvant therapy**			
Radiation	110 (12%)	71 (15%)	0.061
Chemotherapy	103 (11%)	47 (10%)	0.627
**Pre-operative characteristics**			
Re-thoracotomy	73 (8%)	45 (10%)	0.225
Pulmonary infection <4 weeks prior surgery	40 (4%)	35 (7%)	**0.010**
CRP > 3 mg/dl	811 (85%)	427 (91%)	**0.004**
Haemoglobin (mg/dl)	12.7 ± 2.0	12.7 ± 2.0	0.866
pH	7.43 ± 0.04	7.43 ± 0.04	0.272
paO_2_ (mmHg)	76 ± 10	74 ± 14	0.723
paCO_2_ (mmHg)	37 ± 5	37 ± 6	0.240
FEV_1_ < 60%	118 (13%)	100 (22%)	**<0.001**

**Table 2 jcm-11-05774-t002:** Univariate analysis of surgery- and anaesthesia-related characteristics that led to risk factors for PPCs in patients undergoing open thoracotomy lung resection due to primary lung cancer. The data are presented as the number of patients (percentage) or median (interquartile range). PPC = postoperative pulmonary complication; PRBCs = packed red blood cells; OLV = one-lung ventilation; PEEP = positive end-expiratory pressure.

	No PPC (*n* = 954)	PPC (*n* = 472)	*p*-Value
**Surgery-related characteristics**			
Duration of surgery ≥ 195 min	262 (28%)	183 (39%)	**<0.001**
Bronchoplasty	7 (1%)	2 (0%)	0.487
Pneumonectomy	92 (10%)	22 (5%)	**0.001**
Lobectomy	569 (60%)	324 (69%)	**0.001**
Bi-lobectomy	50 (5%)	49 (10%)	**<0.001**
Segment resection	89 (9%)	44 (9%)	0.997
Wedge resection	116 (12%)	19 (4%)	**<0.001**
Intraoperative blood loss	429 ± 508	684 ± 643	**0.001**
Partial ligation of pulmonary artery	119 (13%)	96 (20%)	**<0.001**
**Anaesthesia-related characteristics**			
Total intravenous anaesthesia	430 (45%)	238 (50%)	0.057
Intraoperatively infused crystalloid fluid volume ≥ 6 mL/kg/h	685 (72%)	397 (84%)	**<0.001**
Infused colloids	87 (9%)	47 (10%)	0.610
Intraoperative vasopressors	742 (78%)	385 (82%)	0.098
PRBCs	46 (5%)	41 (9%)	**0.004**
Fresh frozen plasma	12 (1%)	15 (3%)	**0.012**
Continuous epidural or paravertebral analgesia	583 (61%)	249 (53%)	**0.003**
Partial ligation of pulmonary artery	119 (13%)	96 (20%)	**<0.001**
OLV PEEP ≤ 5 cmH_2_O	505 (56%)	286 (64%)	**0.002**
OLV PEEP ≤ 7 cmH_2_O	799 (88%)	390 (88%)	0.851
OLV duration ≥ 175 min	294 (31%)	189 (40%)	**0.001**
No extubation immediately after surgery	31 (3%)	48 (10%)	**<0.001**

## Data Availability

Not applicable.

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
