# Peer review of "Risk Factors for Postoperative Pulmonary Complications Leading to Increased In-Hospital Mortality in Patients Undergoing Thoracotomy for Primary Lung Cancer Resection: A Multicentre Retrospective Cohort Study of the German Thorax Registry"

_jcm, 2022, doi:10.3390/jcm11195774_

Round 1
Reviewer 1 Report
This paper has a good scientific level and it is of interest for anesthesiologist, intensive care physicians or pneumologists. It includes a lot of patients and results, although predictable, have a good statistic significance.
I have one concern regarding the homogeneity among the anesthetic techniques and postoperative control of analgesia between hospitals/centers. The different approaches could have an important impact on postoperative pulmonary complication.
This should be discuss and add at the limitations of this study.
Reviewer 2 Report
Dear authors,
You present a retrospective analysis of patients undergoing OTLR for lung cancer. I believe your study is interesting and well-written. Subsequently, I have but a few comments:
- Why did you choose 195 minutes as a cutoff for length of surgery? This seems like a rather random cutoff time. In your previous study, you used 120 minutes as a cutoff, and for reasons of consistency I would stay with that.
- You note that the „amount of intraoperative blood loss (OR 1.6, 95% CI 1.2-2.1, P=0.001)“ is a significant factor for PPC. What was the cutoff you defined (you must have had a cutoff or a step-wise approach if you present an OR)?
- I would suggest naming the four contributing centers.
- I would suggest to present the p values as „<0.001“ instead of „0.000“ in Figure 1 and the tables. You may also want to consider switching from „1.368“ to „1,368“ etc. in Figure 1.
- You discuss the intriguing pneumonectomy findings. Is it possible that patient selection also plays a role? For pneumonectomy patients, surgeons may apply stricter criteria regarding operability. Patients too unfit for surgery may be offered radio(chemo)therapy instead. Similarly, patients with a high ASA may also be well-selected for surgery.
